# Dietary Tryptophan Levels Impact Growth Performance and Intestinal Microbial Ecology in Weaned Piglets via Tryptophan Metabolites and Intestinal Antimicrobial Peptides

**DOI:** 10.3390/ani11030817

**Published:** 2021-03-14

**Authors:** Zebin Rao, Jinlong Li, Baoshi Shi, Yan Zeng, Yubo Liu, Zhihong Sun, Liuting Wu, Weizhong Sun, Zhiru Tang

**Affiliations:** 1Laboratory for Bio-Feed and Molecular Nutrition, College of Animal Science and Technology, Southwest University, Chongqing 400715, China; raozebin2020@163.com (Z.R.); WSLJL0NG@163.com (J.L.); tto93@163.com (B.S.); sunzh2002cn@aliyun.com (Z.S.); 17752785935@163.com (L.W.); swz2012@swu.edu.cn (W.S.); 2Hunan Institute of Microbiology, Changsha 410009, China; zengyan1977@163.com (Y.Z.); lyubo123@163.com (Y.L.)

**Keywords:** pig, amino acid, indoleamine 2,3-dioxygenase, gut microbial ecology, porcine β-defensin

## Abstract

**Simple Summary:**

Tryptophan (Trp) plays an important role in piglet growth. However, the effect of dietary Trp on microbial flora is still poorly understood. A total of 40 28-d weaned piglets were fed a corn and soybean meal-based diet with 0.14%, 0.21%, 0.28%, or 0.35% Trp for four weeks. The average daily body weight gain, average daily feed intake, feed conversion ratio, spleen index, pancreas index, longissimus dorsi muscle index, plasma insulin, 5-hydroxytryptamine, kynurenine, and Trp concentrations of weaned piglets increased in a Trp dose-dependent manner. Compared with the 0.14% Trp diet, the adequate-Trp diets (0.21%, 0.28%, or 0.35%) down-regulated the relative abundances of 12 genera including *Turicibacter, Prevotella, Mitsuokella, Anaerovibrio, Megasphaera, Succinivibrio, Sutterella, Desulfovibrio*, and *Methanobrevibacter*; up-regulated the abundances of *Ruminococcaceae, Lactobacillus*, and *Muribaculaceae* in the colon; and augmented the mRNA level and concentration of porcine β-defensin 2 in the small intestinal mucosa. Moreover, Trp-adequate diets increased the abundances of Trp hydroxylase, indoleamine 2,3-dioxygenase, porcine β-defensin 2, phosphorylated mammalian target of rapamycin, and phosphorylated protein kinase B in the small intestinal mucosa. In summary, a corn and soybean meal-based diet with 0.35% Trp may be a nutritional strategy to improve growth performance, intestinal mucosal barrier integrity, and intestinal microbial ecology in weaned piglets.

**Abstract:**

Tryptophan (Trp) plays an important role in piglet growth. However, the effect of dietary Trp on microbial flora is still poorly understood. A total of 40 28-d weaned piglets were allocated to four groups with 10 barrows per group and one pig per replicate. Piglets were fed a corn and soybean meal-based diet with 0.14%, 0.21%, 0.28%, or 0.35% Trp for four weeks. Five piglets from each diet group were euthanized, and blood and tissue samples were collected. The average daily body weight gain, average daily feed intake, feed conversion ratio, spleen index, pancreas index, longissimus dorsi muscle index, plasma insulin, 5-hydroxytryptamine, kynurenine, and Trp concentrations of weaned piglets increased in a dose-dependent manner (*p* < 0.05). Compared with the 0.14% Trp diet, the adequate-Trp diets (0.21%, 0.28%, or 0.35%) down-regulated the relative abundances of 12 genera including *Turicibacter, Prevotella, Mitsuokella, Anaerovibrio, Megasphaera, Succinivibrio, Sutterella, Desulfovibrio*, and *Methanobrevibacter* (*p* < 0.05); up-regulated the abundances of *Ruminococcaceae, Lactobacillus*, and *Muribaculaceae* in the colon (*p* < 0.05); and augmented the mRNA level and concentration of porcine β-defensin 2 in the small intestinal mucosa (*p* < 0.05). Moreover, Trp-adequate diets increased the abundances of Trp hydroxylase, indoleamine 2,3-dioxygenase, porcine β-defensin 2, phosphorylated mammalian target of rapamycin, and phosphorylated protein kinase B in the small intestinal mucosa (*p* < 0.05). We noted that a corn and soybean meal-based diet with 0.35% Trp may be a nutritional strategy to improve growth performance, intestinal mucosal barrier integrity, and intestinal microbial ecology in weaned piglets.

## 1. Introduction

Intestinal homeostasis in piglets is impacted by environmental factors (including stress and dietary nutrients) that regulate the flora colonisation. Specifically, weaned stress, an inevitable event for infants and piglets, reconstructs the microbial ecological community in the gastrointestinal tract [1,2]. Tryptophan (Trp) is a vital nutrient for the development of weaned piglets. Trp and its metabolites play important roles in growth performance, intestinal mucosal barrier function, immune regulation, appetite, and general health [3]. Intestinal antimicrobial peptides are important components of the intestinal mucosal chemical barrier and play a key role in defense against intestinal pathogenic microbial infection. In recent years, studies have shown that Trp promotes the expression of β-defensin 2 (BD-2) in the intestinal mucosa [4,5,6,7,8], thereby limiting the negative effects of pathogenic *E. coli* on piglet growth [9]. Our previous study showed that Trp promotes BD-2 expression via the mammalian target of rapamycin (mTOR) pathway and that its metabolite (kynurenine) binds to the aryl hydrocarbon receptor in the rat intestine [10]. Dietary Trp supplementation improves the growth of weanling piglets and regulates the composition of the microbiota in the hindgut [11]. However, it remains unclear how adequate and inadequate dietary Trp impact intestinal microbial ecology. At present, the Trp requirement for 28-day-old weaned piglets recommended by the National Research Council (NRC) is 0.21%. Therefore, a corn- and soybean-based diet containing 0.14% Trp was supplemented with 0.0% (deficient), 0.07% (adequate), 0.14% (excess), or 0.21% Trp (excess) according to the nutritional requirements for weaned piglets in the present study. The purpose of our study was to reveal the mechanism of the effects of insufficient, adequate, and excess Trp on the intestinal microflora and growth performance of weaned piglets.

## 2. Materials and Methods

### 2.1. Animals, Experimental Design, and Diets

Forty 28-d Duroc × Landrace × Yorkshire weaned piglets (5.42 ± 1.52 kg) obtained from the Chongqing Academy of Animal Science (Chongqing, China) and randomly divided into four groups with 10 barrows per group and one pig per replicate. They were fed a corn- and soybean-based diet (containing 0.14% Trp) supplemented with 0.0% (deficient), 0.07% (adequate), 0.14% (excess), or 0.21% Trp (excess). The final Trp levels of the four maize-soybean meal diets were 0.14%, 0.21%, 0.28%, and 0.35%. The experiment lasted for 4 weeks. The ingredients and composition of the basal diet were formulated according to the NRC requirements (2012) and are listed in Table 1. The piglets were kept individually in pens (1.5 m length × 0.5 m width × 0.8 m depth) in a mechanically ventilated and temperature-controlled room (22 ± 1.2 °C). Feed and water were provided on an ad libitum basis. All experimental procedures involving piglets were approved by the License of Experimental Animals (SYXK 2014-0002) of the Animal Experimentation Ethics Committee of Southwest University, Chongqing, China.

### 2.2. Measurements and Sampling

The piglets were weighed on day 28 before the morning feeding. Feed intake was recorded daily throughout the experimental period. Before the morning feeding on day 28, a 10-mL blood sample was collected from the orbital sinus of five piglets selected from each treatment group. The blood samples were left undisturbed for 60 min and then centrifuged at 3500 g for 10 min at 4 °C to harvest the serum. The serum was stored at –20 °C for biochemical analysis and ELISA. After blood sampling, the five piglets selected from each treatment were anaesthetized with an intravenous injection of sodium pentobarbital (50 mg/kg BW) and exsanguinated. The jejunal and ileal mucosa were rinsed with cold saline, then scraped gently with a scalpel blade and collected. The harvested jejunal and ileal mucosa were immediately frozen in liquid N2 and stored at −80 °C for quantitative PCR analysis. Samples of caecum contents collected for analysis of 16S rDNA genes sequencing.

### 2.3. Growth Performance and Diarrhoea Incidence

Feed intake in pigs was recorded daily. The pigs were weighed on day 28 before the morning feeding. The average daily weight gain (ADG), average daily feed intake (ADFI), and feed conversion rate (FCR) were calculated according to the following formulas:ADG (g/d) = (final weight − initial weight) (g)/28 (d),(1)
ADFI (g/d) = total food intake (g)/28 (d),(2)
FCR (g/g) = ADG (g/d)/average daily food intake (g/d)(3)

The diarrhoea incidence, stool shape, and diarrhoea index in pigs were recorded daily according to the scoring criteria reported by Kelly (1990): 0 points for normal faces, 1 point for soft faeces, 2 points for unshaped loose faeces, and 3 points for watery faeces, mucous stools, or purulent bloody stools. The diarrhoea rate and index were calculated according to the following formulas:Diarrhoea rate (%) = number of pigs with diarrhoea/(number of pigs × test days) × 100%,(4)
Diarrhoea index = Sum of diarrhoea scores of each group during the trial period/(number of pigs × test days),(5)

### 2.4. Biochemical and ELISA Analysis

The jejunum mucosa sample (1 g) was homogenized in 9 mL 0.9% NaCl with a polytron (Brinkmann Instruments Inc., Westbury, NY, USA) and centrifuged at 6500 g for 20 min at 4 °C. The supernatant was collected and stored at –80 °C for later analysis. The concentration of growth hormone (GH), cortisol, insulin, melatonin, TPH (Trp hydroxylase), 5-HT (5-hydroxytryptamine), indoleamine 2,3-dioxygenase (IDO), interleukin (IL)-17 and IL-22 in the jejunum and ileum mucosa were determined by porcine-specific ELISA kits (Nanjing Jiancheng Bioengineering Institute, Nanjing, China).

### 2.5. H&E Staining

The morphology of the jejunum and the ileum was analyzed according to the H&E staining method described by Wang et al. [12]. The sliced sample was viewed under an optical microscope (Carl Zeiss Inc., Oberkochen, Bayern, Germany). Digital images were taken using a color video camera (Sony 3CCD-VX3 camcorder) for goblet cell measurements and lymphocyte counts. The villus height and crypt depth were measured using image analysis software (Intronic GmbH & Co., Rothenstein, Berlin, Germany).

### 2.6. Determination of Serum Trp and Kynurenine Concentration

To obtain the supernatant, the collected blood samples were diluted twice with 4% sulfosalicylic acid and centrifuged at 16,000 rpm for 2 min at 4 °C. After the supernatants were filtered with a 0.22 nm filter, the supernatant samples were measured on a Hitachi L-8800 automatic amino acid analyzer with a C_18_ column (250 mm × 4.6 mm, i.d. 5 μm) and a mobile phase including 0.2 M zinc acetate, 8.3 mM acetic acid and 2.8% acetonitrile at a 1.5 mL/min isocratic flow rate. The Trp excitation wavelength was 254 nm and the emission wavelength was 404 nm. The kynurenine excitation wavelength was 365 nm and the kynurenine emission wavelength was 480 nm. The column temperature was 25 °C and the analysis period for each sample was 60 min. A standard curve was created according to the corresponding peak areas of standard samples (Trp 0, 0.625, 1.25, 2.5, 5, 10, and 20 μM; kynurenine 0, 0.05, 0.5, 2, 10, 20, and 50 μM). The contents of the corresponding amino acids in the samples were calculated using this standard curve.

### 2.7. Real-Time RT-PCR

The frozen jejunal and ileal mucosal samples (0.05 g) were homogenized in 5 mL of TRIzol reagent containing RNAlater, and total RNA was isolated according to the manufacturer’s instructions (Invitrogen, Carlsbad, CA, USA). Total RNA was reverse-transcribed to cDNA by an AMV First Strand cDNA Synthesis Kit (Bio Basic Inc., Markham, ON, Canada). Real-time PCR was performed to quantify the relative amounts of glyceraldehyde-3-phosphate (GAPDH) and BD-2 in the jejunal and ileal mucosa. The primers used to amplify genes in the jejunal mucosa are shown in Table 2. PCR analysis was performed using the SYBR Green method on the Applied Biosystems 7900HT Fast Real-Time PCR System (Applied Biosystems, Foster City, CA, USA). The thermal cycling parameters were set as follows: 94 °C for 30 s, followed by 40 cycles of 94 °C for 5 s, annealing (GAPDH, 58 °C; BD-2, 57 °C) for 20 s, followed by 72 °C for 20 s. The PCR products were identified by melting curve analysis and sequencing (Sangon Biotech, Shanghai, China). Standard calibration curves for the target genes were created according to the cDNA concentration and Ct value. The expression levels of the target genes were estimated by the calibration curve and normalized against GAPDH expression using REST 2009 Software (Qiagen, Düsseldorf, Germany).

### 2.8. Western Blot

Samples of jejunal and ileal mucosa (100 mg) were homogenised in 1 mL of RIPA buffer (1.0 mM ethylenediamine tetraacetic acid (EDTA)), 50 mM Tris-base, 0.1% sodium dodecyl sulfate (SDS), 150 mM NaCl, 1% Triton X-100, 1% sodium deoxycholate, and 1 mM phenylmethanesulfonyl fluoride (PMSF) and separated by SDS-polyacrylamide gel electrophoresis. The proteins were transferred to a polyvinylidene fluoride (PVDF) membrane by the semi-dry transfer method. The PVDF membranes were blocked in a blocking buffer overnight at 4 °C, then incubated in blocking buffer with rabbits antibodies for β-actin (13E5) (5125S, 1:1000, Cell Signaling Technology, Boston, MA, USA), BD-2 (ab178728, 1:1000, Abcam, Cambridge, MA, USA), The mechanistic target of rapamycin complex 1 (mTORC1) (ab32028; 1:1000, Abcam), Ser2448-phosphorylated mTOR (ab109268, 1:1000, Abcam), protein kinase B (AKT) (9272, 1:1000, Cell Signaling Technology, Boston, MA, USA), Ser473-phosphorylated AKT (FAB127472; 1:1000; Fantibody). They were then incubated in blocking buffer with F(ab)2 of goat anti-rabbit Ig (1:2500) labelled with horseradish peroxidase diluted in PBSS. The PVDF membrane was soaked in a chemiluminescent liquid (Millipore, Boston, MA, USA). Pictures were taken using a Chemiluminescence Imaging System (Bio-Rad, Bio rad Hercules, CA, USA).

### 2.9. Analysis of 16S rDNA Genes Sequencing

The DNA samples were extracted from Samples of caecum contents using the Power Fecal DNA Isolation Kit (Mobio, Carlsbad, CA, USA), stained using the Quant-iT Pico Green dsDNA Kit (Invitrogen Ltd., Paisley, UK) and quantified using a Nanodrop spectrophotometer (Nyxor Biotech, Paris, France).

For DNA MiSeq sequencing, PCR amplification of the V4 region of bacterial 16S rDNA was performed using the universal primers 515F (5ʹ-GTGCCAGCM GCCGCGGTAA-3ʹ) and 806R (5ʹ-GGACTACHVGGGTWTCTAAT-3ʹ), incorporating the FLX Titanium adapters and a sample barcode sequence. The cycling parameters were as follows: 5 min of initial denaturation at 95 °C; 25 cycles of denaturation at 95 °C (30 s), annealing at 55 °C (30 s), and elongation at 72 °C (30 s); and final extension at 72 °C for 5 min. Three separate PCR reactions for each sample were pooled for MiSeq sequencing. The PCR products were separated by 1.5% agarose gel electrophoresis and purified using the QIAquick Gel extraction kit (QIAGEN, Düsseldorf, Germany). Amplicons were quantified using the Quant-iT Pico Green dsDNA Assay Kit (Invitrogen). Equal concentrations of amplicons were pooled from each sample. Libraries were constructed using the TruSeq DNA PCR-Free Sample Prep Kit and MiSeq sequencing was performed with the MiSeq Reagent Kit v2 (Illumina, San Diego, CA, USA).

Raw reads were obtained from MiSeq sequencing for analysis. All reads were aligned according to barcode and primer sequences. The resulting sequences were further screened and filtered for quality and length. Sequences that were less than 200 nucleotides or contained ambiguous characters, over two primer mismatches, or mononucleotide repeats of over six nucleotides were removed. The high-quality sequences were assigned to samples according to the barcodes. Sequences with a similarity level of at least 97% were clustered into OTUs (operational taxonomic units) using UPARSE algorithm 7. The representative sequences at an OTU distance of 0.03 were obtained and classified using Ribosomal Database Project (RDP) classifier. Any sequences with annotations for chloroplast, mitochondrial, or archaeal OTUs not identified as being of bacterial origin were excluded from all further analyses. We calculated the abundance-based coverage estimator (ACE), bias-corrected Chao1 richness estimator, and Shannon and Simpson diversity indices. All analyses were performed using mothur (v1.24) (http://www.mothur.org, accessed on 19 May 2020). Based on the above results for community composition (compositional) and development level and system level (phylogenetic), the analysis of species-level differences, correlation, and alpha, beta, and community structure diversity was performed using QIIME [13], mothur [14], and R 4.0.3 software (R Foundation for Statistical Computing, Vienna, Austria).

### 2.10. Statistical Analysis

All data are presented as means ± Standard Error of Mean (SEM). The data were subjected to one-way analysis of variance using the general linear model procedure in the SAS 8.2statistical software (SAS Institute, Inc., Cary, NC, USA), according to a completely randomised one-factorial design. The LSD test was performed to identify differences between groups. Significance was set at *p* < 0.05.

## 3. Results

### 3.1. Growth Performance and Organ and Tissue Development

As shown in Table 3, dietary Trp significantly changed the final weight, ADG, AFDI, and FCR in piglets. Specifically, an increase was observed in ADG, ADFI, and FCR of weaned piglets in response to increased dietary Trp levels (*p* < 0.05). Dietary Trp levels of 0.21%, 0.28%, and 0.35% decreased the diarrhoea rate (36.4%, 34.7%, and 33.8% vs. 43.5%) and diarrhoea index (0.32, 0.28, and 0.26 vs. 0.56) of weaned piglets compared with those of 0.14% dietary Trp. The indices of the spleen, pancreas, and longissimus dorsi muscle were also significantly increased by dietary Trp. Specifically, the piglets fed the diets with 0.28% and 0.35% Trp exhibited increased thyroid and mesenteric lymph node weights and longissimus dorsi muscle index when compared with those of piglets fed the diet with 0.14% Trp. The weights of the thymus, spleen, and pancreas and the spleen index of weaned piglets in the 0.28% and 0.35% Trp groups were higher than those of piglets in the 0.14% and 0.21% Trp groups. The liver weight and jejunum and ileum lengths of piglets in the 0.28% and 0.35% Trp groups were higher than those of piglets in the 0.14% Trp group. The jejunum weight of piglets in the 0.28% and 0.35% Trp groups was higher than that of piglets in the 0.14% and 0.21% Trp groups, and the ileum weight of piglets in the 0.28% and 0.35% Trp groups was higher than that of piglets in the 0.14% Trp group.

### 3.2. Morphology of the Jejunum, Ileum and Colon Mucosa

The results for intestinal mucosa morphology of the piglets are presented in Table 4 and Figure 1. Dietary Trp levels significantly changed the villus height and villus height/crypt depth in the jejunum. Dietary Trp levels significantly changed the villus height and villus height/crypt depth in the ileum. Specifically, piglets fed the diet with 0.21% Trp exhibited increased villus height and ratio of villus height to crypt depth in the jejunum and ileum when compared with those of the piglets fed the diet with 0.14% Trp. When compared with the piglets fed the 0.14% and 0.21% Trp diets, the crypt depth, villus height, and ratio of villus height to crypt depth in the jejunum and ileum was increased in piglets fed the 0.28% and 0.35% Trp diets. There were no differences in the villus height, crypt depth, or ratio of villus height to crypt depth in the jejunum and ileum of piglets between the 0.28% Trp group and the 0.35% Trp group. There were no differences in the villus height, crypt depth, or ratio of villus height to crypt depth in the colon or crypt depth in the jejunum and ileum of piglets among the four groups.

### 3.3. Concentrations of Hormones and Tryptophan Metabolic Products in Serum and Activities of Tryptophan Hydroxylase and Indoleamine 2,3-Dioxygenase in the Jejunal and Ileal Mucosa

The results for hormones and Trp metabolic products are presented in Table 5. In general, dietary Trp levels significantly changed the concentrations of GH, insulin, melatonin, kynurenine, and Trp in the plasma. Trp also significantly affected the enzymatic activities of TPH and IDO in the jejunal mucosa, TPH and IDO activity in the ileal mucosa, and concentrations of 5-HT and TPH in the brains of piglets. Specifically, the serum insulin concentration of piglets in the 0.28% and 0.35% Trp groups was higher than that of piglets in the 0.14% and 0.21% Trp groups. The enzyme activity of TPH in the jejunal mucosa of piglets was increased in response to increased dietary Trp. The enzyme activity of IDO in the jejunal and ileal mucosa of piglets in the 0.21%, 0.28%, and 0.35% Trp groups was higher than that of piglets in the 0.14% Trp group. The kynurenine concentration in the plasma of piglets was increased in response to increased dietary Trp. When compared with piglets fed the 0.14% Trp diet, the serum Trp concentration was increased in piglets fed the 0.21%, 0.28%, and 0.35% Trp diets.

### 3.4. Concentrations of Interleukin 17, Interleukin 22 in the Jejunal and Ileal Mucosa of Weaned Piglets

The results for pro-inflammatory factors are presented in Table 6. The dietary Trp level significantly changed the concentrations of interleukin 17 (IL-17) in the jejunal mucosa and the concentrations of IL-17 and interleukin 22 (IL-22) in the ileal mucosa of piglets. Specifically, the concentrations of IL-22 in the jejunal mucosa of piglets fed the 0.28% and 0.35% Trp diets were lower than those of piglets fed the 0.14% Trp diet. When compared with piglets fed the 0.14% and 0.21% Trp diets, the concentration of IL-17 in the jejunal mucosa was decreased in piglets fed the 0.28% and 0.35% Trp diets. Piglets fed the 0.28% and 0.35% Trp diets exhibited a decreased concentration of IL-17 in the ileal mucosa when compared with that of piglets fed the 0.14% and 0.21% Trp diets. The concentration of IL-22 in the ileal mucosa of piglets fed the 0.14% Trp diet was lower than that of piglets fed the 0.21%, 0.28%, and 0.35% Trp diets.

### 3.5. The mRNA Levels of Porcine β-Defensin 2 Concentrations in the Jejunal and Ileal Mucosa

The results for mRNA levels are presented in Table 7. Dietary Trp significantly changed the mRNA levels of porcine BD-2 in the jejunal mucosa and the ileal mucosa of piglets. Specifically, the mRNA level of porcine BD-2 in the jejunal and ileal mucosa of piglets fed the 0.14% Trp diet was lower than that of piglets fed the 0.21%, 0.28%, and 0.35% Trp diets.

### 3.6. Protein Levels of Porcine β-Defensin 2, Mammalian Target of Rapamycin, Phosphorylated Mammalian Target of Rapamycin, Protein Kinase B and Phorylated Protein Kinase B Concentrations in the Jejunal and Ileal Mucosa

The results for p-mTOR, p-AKT, and porcine BD-2 concentrations are presented in Figure 2. In general, dietary Trp significantly changed the protein concentrations of p-mTOR, p-AKT, and porcine BD-2 in the colon mucosa of weaned piglets. Specifically, the relative concentrations of p-mTOR in the colon mucosa of weaned piglets fed the 0.14% and 0.21% Trp diets were lower than those of piglets fed the 0.35% Trp diet. When compared with piglets fed the 0.14% and 0.21% Trp diets, piglets in the 0.28% and 0.35% Trp diets group exhibited increased relative protein concentrations of p-AKT and porcine BD-2 in the colon mucosa.

### 3.7. Intestinal Microbial Flora

A Venn diagram was used to explore similarities and differences in microbial communities among groups, showing that the intestinal microbial communities in the caecum contents of piglets in the four groups had 747 common OTUs, accounting for 68%, 64.9%, 64%, and 60.7% of OTUs in the 0.14%, 0.21%, 0.28%, and 35% Trp groups, respectively (Figure 3A). The results of the weighted UniFrac clustering analysis showed that the difference in intestinal microbial flora from large to small is in the following order: 0.14%, 0.21%, 0.35%, and 0.28% Trp (Figure 3B). The results of the weighted UniFrac PCoA analysis showed that the similarity of intestinal microbial flora from large to small is in the following order: 0.14%, 0.35%, 0.21% and 0.28% (Figure 3C).

As shown in Table 8, more than 90% of the bacteria in the colonic contents belonged to Firmicutes, Bacteroidetes, and Proteobacteria. In general, dietary Trp significantly changed the coverage estimator (ACE), Chao1 and PD. Specifically, ACE, Chao1, and PD of piglets fed the 21%, 0.28%, and 0.35% Trp diets were higher than those of piglets fed the 0.14% Trp diet. ACE and Chao1 of piglets fed the 0.35% Trp diet were higher than those of piglets fed the 0.14%, 21%, and 0.28% Trp diets. Dietary Trp significantly increased the relative abundance of Bacteroidetes, Tenericutes, Verrucomicrobia in the caecum microbiota of weaned piglets. Dietary Trp significantly decreased the relative abundance of Firmicutes, Proteobacteria; Euryarchaeota in the caecum microbiota of weaned piglets. Dietary Trp did not significantly change the relative abundance of Actinobacteria, Spirochaetes, or Cyanobacteria.

As shown in Table 9, in the caecum microbiota of weaned piglets, dietary Trp significantly increased the relative abundance of *Phascolarctobacterium*, *Succiniclasticum*, and *Dialister* in Veillonellaceae, *Blautia*, *Roseburia*, uncultured species, *Lachnospira*, and *Coprococcus* in Lachnospiraceae, *Faecalibacterium*, *Ruminococcus*, *Oscillospira*, and uncultured species in Ruminococcaceae, *Streptococcus* in Streptococcaceae, *Lactobacillus* in Lactobacillaceae, uncultured species in Bacteroidales, uncultured species in S24-7, CF231 in Paraprevotellaceae, *Treponema* in Spirochaetaceae, uncultured species in RFP12, and *Akkermansia* in Verrucomicrobiaceae.

In the caecum microbiota of weaned piglets, dietary Trp significantly decreased the relative abundance of *Turicibacter* in Turicibacteraceae, *Prevotella* in Prevotellaceae, uncultured species, *Mitsuokella*, *Anaerovibrio*, and *Megasphaera* in Veillonellaceae, uncultured species in Enterobacteriaceae, *Succinivibrio* in Succinivibrionaceae, uncultured species in Rhodocyclaceae, Sutterella in Alcaligenaceae, *Desulfovibrio* in Desulfovibrionaceae, and *Methanobrevibacter* in Methanobacteriaceae.

## 4. Discussion

### 4.1. Growth Performance, Organ Index, and Intestinal Mucosal Morphology

At present, the Trp requirement of 28-day weaned piglets recommended by the NRC (2012) is 0.21%. In the present study, a corn- and soybean-based diet was formulated according to the nutritional requirements for weaned piglets. The basal diet contained 0.14% Trp and was supplemented with 0.0% (deficient; 0.14% group), 0.07% (adequate; 0.21% group), 0.14% (excess; 0.28% group), or 0.21% Trp (excess; 0.35% group). We found that there was a linear increase in the final weight, ADG, ADFI, and FCR of weaned piglets with increasing dietary Trp level; poor growth performance emerged in the 0.14% Trp group and 0.35% dietary Trp maximised growth performance. Trp is an indispensable amino acid for pig growth. In recent years, studies have noted that Trp affects piglet growth performance mainly by regulating appetite and feed intake [15,16]. Shen et al. [17] also reported that piglets obtained maximised growth performance with 0.8% Trp. Piglets can detect changes in metabolism caused by dietary Trp deficiency and respond with an aversion against the Trp-deficient diet [18].

Dietary Trp deficiency directly decreases the concentration of Trp in the brain and then reduces 5-HT synthesis in the hypothalamus. We found that Trp deficiency decreases serum GH, insulin, melatonin, kynurenine, and Trp concentration, as well as the concentrations of 5-HT and TPH in the brains of piglets in the 0.14% Trp group. We found that adequate and excess Trp in the diet increased serum GH, insulin, melatonin, kynurenine, and Trp concentration, and the concentrations of 5-HT and TPH in the brains of piglets. There is evidence that dietary supplementation with Trp can achieve improving growth performance and stress adaptation for piglets, resulting from increased concentrations of hypothalamic 5-HT and reduced secretion of stress hormones [19]. Organ index is a classic parameter to measure growth development and immune function in animals. In the present study, we observed boosted spleen index associated with increasing levels of Trp, as well as weight index and length index of the ileum and length index of the jejunum. Pertsov et al. [20] reported that melatonin (a metabolite of Trp) increases the thymus index and spleen index under normal conditions and prevents thymus atrophy induced by stress. Therefore, Trp and its metabolites play an important role in promoting growth performance and organ development.

Consistently, we found that the villus height and the ratio of villus height to crypt depth in the 0.21%, 0.28%, and 0.35% Trp groups were higher than those in the 0.14% Trp group. We also found that the relative protein concentrations of p-mTOR and p-AKT in the 0.28% and 0.35% Trp groups were higher than those in the 0.21% and 0.14% Trp groups. This study indicates that an appropriate dose of Trp was beneficial to both the activation of mTOR cell signalling and enhancement of intestinal mucosal growth. Koopmans et al. [21] found that piglets fed with high levels of Trp had increased intestinal villus height, an increased ratio of the villus height to the depth of the recess in the whole small intestine, and a tendency for reduced depth of the recess. A possible cause proposed by Marion et al. is the sudden change in the piglets’ diet structure after weaning and the influence of other factors which lead to insufficient nutrient intake, which hinders intestinal mucosal development [22]. In studies by Tossou et al. [23], weaned piglets provided a high level of dietary Trp had increased depth of the intestinal recess and a reduced ratio of villus height to depth of the recess.

### 4.2. Diarrhoea Rate, Intestinal Microbial Ecology, and Intestinal Antimicrobial Peptides

Host defence peptides exert both antimicrobial and immunomodulatory activities and contribute to epithelial immune defence [24]. There were correlations between dietary Trp level, the expression of antimicrobial peptides, intestinal microbial ecology, and diarrhoea [25]. In this experiment, the bacteria in the colonic contents of the four test groups were mainly of Proteobacteria, Firmicutes, and Bacteroidetes, and the total amount of microbes of these three phyla reached more than 90%. The total amount of microbes of Verrucomicrobia, Spirochaetes, and Euryarchaeota was less than 10%. It was found that the dominant bacteria of intestinal microflora in weaned piglets are mainly Proteobacteria, Firmicutes, and Bacteroidetes [26]. The result that sufficient and excess dietary Trp decreased the diarrhoea rate and diarrhoea index, compared with the levels with deficient dietary Trp, is related to the fact that sufficient and excess dietary Trp increased the abundance of Firmicutes, Proteobacteria, and Euryarchaeota and decreased the abundance of Verrucomicrobia and Bacteroidetes, compared with the levels with deficient dietary Trp.

We also found that adequate dietary Trp significantly increased the relative abundance of 20 genera (*Phascolarctobacterium, Succiniclasticum*, and *Dialister* in Veillonellaceae; *Blautia, Roseburia*, uncultured species, *Lachnospira*, and *Coprococcus* in Lachnospiraceae; *Faecalibacterium, Ruminococcus, Oscillospira*, and uncultured species in Ruminococcaceae; *Streptococcus* in Streptococcaceae; *Lactobacillus* in Lactobacillaceae; uncultured species in Bacteroidales; uncultured species in S24-7; CF231 in Paraprevotellaceae; *Treponema* in Spirochaetaceae; uncultured species in RFP12; and *Akkermansia* in Verrucomicrobiaceae) in the caecum microbiota of weaned piglets. Dietary Trp significantly decreased the relative abundance of 12 genera (*Turicibacter* in Turicibacteraceae; *Prevotella* in Prevotellaceae; uncultured species, *Mitsuokella, Anaerovibrio*, and *Megasphaera* in Veillonellaceae; uncultured species in Enterobacteriaceae; *Succinivibrio* in Succinivibrionaceae; uncultured species in Rhodocyclaceae; *Sutterella* in Alcaligenaceae; *Desulfovibrio* in Desulfovibrionaceae; and *Methanobrevibacter* in Methanobacteriaceae) in the caecum microbiota of weaned piglets. The fact that dietary Trp significantly changes intestinal microbial ecology is related to the enhancement of intestinal antimicrobial peptides by high dietary levels of Trp (0.28% and 0.35%).

Dietary Trp regulate the composition of the microbiota in the hindgut and front-gut, and up-regulate porcine BD-2 [8,11]. The fact that dietary Trp significantly changes the levels of 32 genera and does not significantly affect the levels of 14 genera is related to the fact that high levels of dietary Trp contribute to enhanced inflammatory cytokine (TNF-α, IL-17 and IL-22) concentrations in the brain and mRNA levels of porcine BD-2 in the jejunal and ileal mucosa of piglets. Cytokines are crucial for immune and inflammatory responses. Porcine BD-2 is a major group of porcine antimicrobial peptides that plays an important role in both mucosal barrier function and immune response due to their antimicrobial, chemotactic, and regulatory activities [27,28]. Weaning stress and pathogen infection are associated with reduced expression of porcine BD-2 [2,29]. The exogenous supply of porcine BD-2 improves the intestinal integrity and growth performance of weaned piglets [30]. One study found that IL-17 and IL-22 cooperatively enhance the expression of antimicrobial peptides in keratinocytes [31]. We also found that the relative protein concentrations of p-mTOR and p-AKT in the 0.28% and 0.35% Trp groups were higher than those in the 0.21% and 0.14% Trp groups, indicating that an appropriate dosage of Trp was favourable for both the activation of mTOR cell signalling and the expression of intestinal antimicrobial peptides. Hashimoto et al. [4] found that the expression of AKT protein and mTOR protein in intestinal mucosa increased with increasing Trp levels in the diet. Wang et al. [32] showed that an increase in Trp concentration significantly increased the level of mTOR protein in intestinal epithelial cells. Schiering et al. [33] proposed that the canine Trp metabolite, urinary ammonia, can regulate immune cells to produce cytokines (IL-17, IL-22) by acting on the aryl hydrocarbon receptor (AhR) in immune cells and induce intestinal epithelial cells to secrete antimicrobial peptides. Our previous study showed that Trp promotes BD-2 expression via the mTOR pathway and its metabolite (kynurenine) bonding to the aryl hydrocarbon receptor in rat intestine [10].

## 5. Conclusions

In conclusion, the results of the present study indicate that adequate provision of Trp decreases the diarrhoea rate and enhances growth performance, intestinal mucosal growth, and microbial ecology in the hindgut as shown by the enhanced abundances of ADG and FCR, as well as the up-regulation of intestinal antimicrobial peptides (porcine BD-2). These beneficial effects of Trp are associated with Trp metabolites, the activation of mTOR signalling, and the enrichment of probiotics in the small intestine of weaned pigs. A corn and soybean meal-based diet with 0.35% Trp may be a nutritional strategy to improve growth performance, intestinal mucosal barrier integrity, and intestinal microbial ecology in weaned piglets.

## Figures and Tables

**Figure 1 animals-11-00817-f001:**
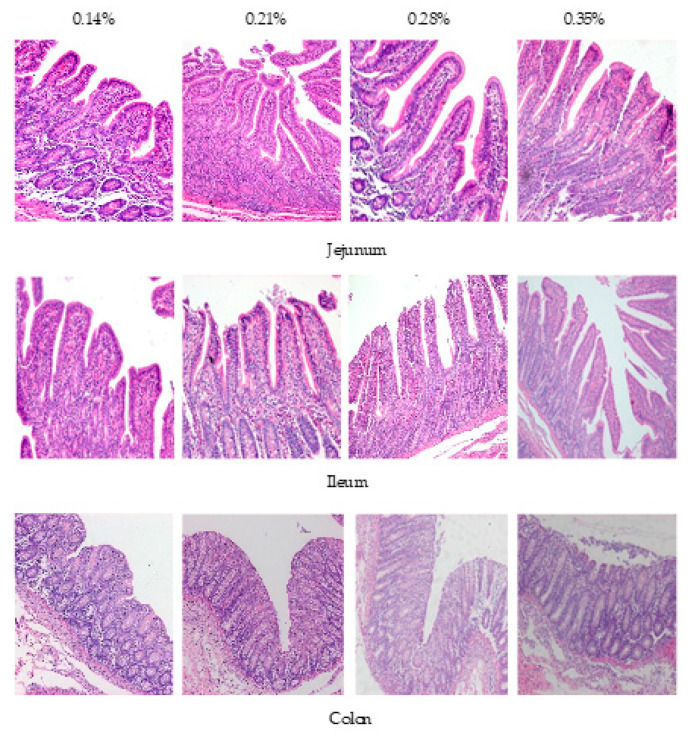
Effects of dietary tryptophan levels on the mucosal morphology of the jejunum, ileum, and colon of weaned piglets (400×).

**Figure 2 animals-11-00817-f002:**
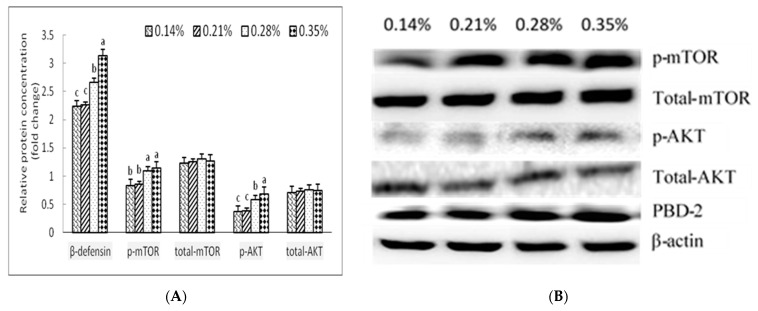
Effects of dietary tryptophan levels on the relative protein concentrations of p-mTOR, mTOR, p-AKT, AKT, and PBD-2 in the colon mucosa of weaned piglets (*n* = 5). (**A**) The results of western blot analysis. (**B**) Western blot figure. a, b, c: values with different letter superscripts within the same index mean significant differences (*p* < 0.05). All data are mean ± SEM. Abbreviations: p-mTOR, phosphorylated mammal target of rapamycin; p-AKT, phosphorylated protein kinase B; PBD-2, β-defensin 2.

**Figure 3 animals-11-00817-f003:**
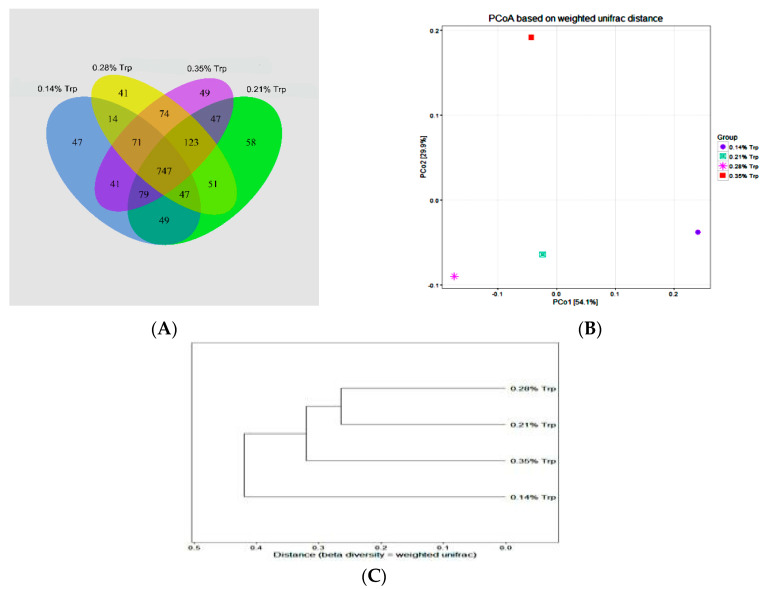
Effect of dietary tryptophan levels on the bacterial community in the colon of weaned piglets. (**A**) Venn diagram comparing the four groups. (**B**) The results of weighted UniFrac PCoA analysis. (**C**) The results of weighted UniFrac clustering analysis. Data are presented as the mean (*n* = 5). Abbreviations: 1, the group supplemented with 0.14% tryptophan; 2, the group supplemented with 0.21% tryptophan; 3, the group supplemented with 0.28% tryptophan; 4, the group supplemented with 0.35% tryptophan.

**Table 1 animals-11-00817-t001:** Ingredients and composition of experiment diets (DM basis).

Ingredients	Treatments			
	0.14%	0.21%	0.28%	0.35%
Corn	68.72	68.65	68.58	68.51
Soybean meal	8.07	8.07	8.07	8.07
Corn gluten meal	8.67	8.67	8.67	8.67
Whey	5.00	5.00	5.00	5.00
Wheat bran	5.00	5.00	5.00	5.00
Monocalcium phosphate	0.44	0.44	0.44	0.44
Limestone	1.00	1.00	1.00	1.00
Salt	0.30	0.30	0.30	0.30
Trace mineral premix ^1^	1.00	1.00	1.00	1.00
Threonine	0.18	0.18	0.18	0.18
Lysine	0.82	0.82	0.82	0.82
Methionine	0.10	0.10	0.10	0.10
Tryptophan ^2^	0.00	0.07	0.14	0.21
Alanine	0.70	0.64	0.58	0.52
Total	100	100	100	100
**Composition**				
DE (MJ/kg) ^3^	14.1	14.1	14.1	14.1
CP (%)	16.2	16.2	16.2	16.2
Ca (%)	0.73	0.73	0.73	0.73
CF (%)	2.60	2.60	2.60	2.60
Available P (%)	0.36	0.36	0.36	0.36
Lysine (%)	1.27	1.27	1.27	1.27
Methionine (%)	0.37	0.37	0.37	0.37
Threonine (%)	0.75	0.75	0.75	0.75
Tryptophan (%)	0.14	0.21	0.28	0.35

^1^ Provided the following per kilogram of diet: 1845 IU vitamin A, 200 IU vitamin D, 12 IU vitamin E, 0.48 mg vitamin K, 9.3 mg pantothenic acid, 3.1 mg riboflavin, 0.29 mg folic acid, 29.0 mg niacin, 1.0 mg thiamine, 4.3 mg vitamin B6, 0.05 mg biotin, 0.016 mg vitamin B12, and 0.43 g choline. Provided the following per kilogram of diet: 84 mg Zn, 97 mg Fe, 3.2 mg Mn, 5.2 mg Cu, 0.14 mg I, and 0.259 mg Se. ^2^
l-tryptophan was purchased from Sangon Biotech (Shanghai, China). ^3^ DE was calculated value and others were measured values of a corn- and soybean-based diet (0.14%). Tryptophan level is calculated in the rest 3 treatment (0.21%, 0.28%, and 0.35%). Abbreviations: DE, digestible energy; CP, crude protein; CF, crude fibre.

**Table 2 animals-11-00817-t002:** Primer sequences.

Gene	Sequence Number	Length of PCR Product (bp)	Primer Sequences (5′→3′)	Tm Value
GAPDH	NM001206359.1	149	F: GAAGGTCGGAGTGAACGGAT R: CATGGGTAGAATCATACTGGAACA	65
PBD-2	NM214442	153	F: GCTGCTGCTGCTGACTGTC R: CAATCCTGTTGAAGAGCGGG	63

**Table 3 animals-11-00817-t003:** Effects of dietary tryptophan levels on growth performance and the morphological development of organs and tissues of weaned piglets.

Items	Dietary Tryptophan Levels				SEM	*p*-Value		
	0.14%	0.21%	0.28%	0.35%		L	Q	C
Initial weight (kg)	5.51	5.42	5.41	5.37	0.27	0.986	0.986	0.986
Final weight (kg)	6.73d	7.80c	8.71b	9.91a	0.29	<0.001	0.821	0.715
ADG (g/d)	43.4d	85.0c	118b	162a	3.53	<0.001	0.678	0.191
ADFI (g/d)	138d	182c	242b	310a	7.75	<0.001	0.126	0.826
FCR (g/g)	0.31 c	0.47b	0.49b	0.52 a	0.05	<0.001	<0.001	<0.001
Diarrhoea rate (%)	43.5	36.4	34.7	33.8				
Diarrhoea index	0.56	0.32	0.28	0.26				
Mesenteric lymph nodes (g)	16.4b	23.9ab	24.4a	30.0a	2.54	0.002	0.716	0.293
Jejunum length (cm)	356c	399bc	482ab	524a	27.9	<0.001	0.978	0.529
Ileum length (cm)	300c	334bc	402ab	433a	23.3	<0.001	0.956	0.512
Thyroid index (g/kg)	0.76	0.76	0.93	0.80	0.10	0.548	0.541	0.320
Thymus index (g/kg)	0.45	0.44	0.57	0.49	0.06	0.378	0.576	0.235
Liver index (g/kg)	26.1	25.9	33.3	29.9	4.29	0.344	0.711	0.345
Spleen index (g/kg)	1.89b	1.96b	2.78a	2.72a	0.29	0.020	0.823	0.22
Pancreas index (g/kg)	1.32b	1.67ab	2.24a	1.96ab	0.21	0.020	0.157	0.263
Longissimus dorsi muscle index (g/kg)	9.38b	12.3ab	14.9a	14.5a	1.55	0.020	0.293	0.683
Jejunum index (g/kg)	25.9	25.4	31.0	32.0	3.44	0.138	0.834	0.490
Ileum index (g/kg)	19.7	19.9	23.8	25.0	2.57	0.104	0.855	0.587

a, b, c, d: Values in the same row with different letter superscripts mean significant differences (*p* < 0.05). Data are presented as the mean ± SEM (*n* = 6). Abbreviations: ADG, average daily body gain; ADFI, average daily feed intake; FCR, feed conversion ratio; L, linear; Q, quadratic; C, cubic.

**Table 4 animals-11-00817-t004:** Effects of dietary tryptophan levels on the morphology of the jejunum, ileum and colon mucosa of weaned piglets.

Items	Dietary Tryptophan Levels				SEM	*p*-Value		
	0.14%	0.21%	0.28%	0.35%		L	Q	C
**Jejunum**								
Villus height (µm)	278c	320b	330a	334a	9.75	0.003	0.028	0.028
Crypt depth (µm)	115	120	117	118	4.52	0.727	0.707	0.497
Villus height/crypt depth	2.42c	2.67b	2.75a	2.83a	0.14	0.031	0.031	0.042
**Ileum**								
Villus height (µm)	239c	247b	261a	268a	7.24	0.003	0.007	0.001
Crypt depth (µm)	94.6	96.1	101	103	5.18	0.064	0.331	0.078
Villus height/crypt depth	2.53c	2.57b	2.58a	2.60a	0.13	0.023	0.012	0.048
**Colon**								
Villus height (µm)	218	227	219	235	10.1	0.793	0.242	0.634
Crypt depth (µm)	49.5	44.0	45.5	42.0	6.54	0.487	0.883	0.690
Villus height/crypt depth	4.23	5.41	6.25	5.08	0.69	0.293	0.111	0.595

a, b, c: Values in the same row with different letter superscripts mean significant differences (*p* < 0.05). Data are presented as the mean ± SEM (*n* = 6). Abbreviations: L, linear; Q, quadratic; C, cubic.

**Table 5 animals-11-00817-t005:** Effects of dietary tryptophan levels on the concentrations of hormones and tryptophan metabolic products in the serum and the enzyme activities of TPH and IDO in the jejunal and ileal mucosa of weaned piglets.

Items	Dietary Tryptophan Levels				SEM	*p*-Value		
	0.14%	0.21%	0.28%	0.35%		L	Q	C
**Serum** (ng/mL)								
GH	2.19c	2.32b	2.42a	2.44a	0.12	0.003	0.043	0.045
Cortisol	246	229	243	266	19.3	0.880	0.135	0.647
Insulin	25.6b	28.3b	30.9a	30.8a	1.09	0.002	0.207	0.612
Melatonin	54.8c	58.9b	60.9a	61.3a	1.18	0.012	0.021	0.402
Kynurenine	0.06d	8.55c	14.6b	30.6a	0.22	<0.001	0.001	0.162
Tryptophan	0.71c	0.99b	2.23a	2.26a	0.03	<0.001	<0.001	<0.001
**Jejunum mucosa** (ng/g protein)								
TPH	4.16d	5.40c	9.51b	12.8a	0.16	0.038	0.253	0.884
IDO	28.2b	32.6a	33.1a	33.0a	0.61	<0.001	<0.001	<0.001
**Ileum mucosa** (ng/g protein)								
TPH	2.61c	2.75bc	3.33b	4.34a	0.22	<0.001	0.002	0.237
IDO	30.2b	32.7a	34.3a	34.2a	0.55	<0.001	0.065	0.998
**Brain** (ng/g protein)								
5-HT	108c	129b	148a	147a	2.73	<0.001	0.032	0.759
TPH	22.41c	28.42b	31.52a	31.01a	0.633	<0.01	<0.01	<0.01

a, b, c, d: Values in the same row with different letter superscripts mean significant differences (*p* < 0.05). Data are presented as the mean ± SEM (*n* = 6). Abbreviations: GH, growth hormone; TPH, tryptophan hydroxylase; IDO; indoleamine 2,3-dioxygenase; 5-HT, 5-hydroxytryptamine; L, linear; Q, quadratic; C, cubic.

**Table 6 animals-11-00817-t006:** Effect of dietary tryptophan levels on the concentrations of TNF-α, IL-17, and IL-22 in the jejunal and ileal mucosa of weaned piglets.

Items	Dietary Tryptophan Levels				SEM	*p*-Value		
	0.14%	0.21%	0.28%	0.35%		L	Q	C
**Jejunum mucosa (ng/g protein)**								
IL-17	90.6c	120b	139a	146a	2.71	<0.001	<0.001	<0.001
IL-22	60.4b	60.2b	63.2a	70.4a	3.07	<0.001	<0.001	<0.001
**Ileum mucosa (ng/g protein)**								
IL-17	100c	132b	151a	155a	3.02	<0.001	0.003	<0.001
IL-22	61.3b	64.0b	64.2b	71.6a	2.35	0.025	0.048	0.925

a, b, c: Values in the same row with different letter superscripts mean significant differences (*p* < 0.05). Data are presented as the mean ± SEM (*n* = 6). Abbreviations: IL-17, interleukin 17; IL-22, interleukin 22; L, linear; Q, quadratic; C, cubic.

**Table 7 animals-11-00817-t007:** Effect of dietary tryptophan levels on the mRNA level of PBD-2 in the jejunal and ileal mucosa in weaned piglets.

Items	Dietary Tryptophan Levels				SEM	*p*-Value		
	0.14%	0.21%	0.28%	0.35%		L	Q	C
**Jejunum mucosa**								
PBD-2	0.60c	0.93b	1.13b	1.76a	0.11	<0.001	0.196	0.249
**Ileum mucosa**								
PBD-2	0.47c	1.03b	1.44a	1.77a	0.12	<0.001	0.371	0.895

a, b, c: Values in the same row with different letter superscripts mean significant differences (*p* < 0.05). Data are presented as the mean ± SEM (*n* = 6). Abbreviations: PBD-2, porcine β-defensin 2; L, linear; Q, quadratic; C, cubic.

**Table 8 animals-11-00817-t008:** Effect of dietary tryptophan levels on coverage estimator (ACE), Chao1, Shannon, Simpson, and PD diversity indices and the microbial composition at the phylum level in the caecum microbiota of weaned piglets.

Items	Dietary Tryptophan Levels				SEM	*p*-Value		
	0.14%	0.21%	0.28%	0.35%		L	Q	C
Observed	407.8	439.6	449.2	469.2	25.45	0.11	0.82	0.78
ACE	558c	627b	634b	677a	29.08	0.01	0.2	0.55
Chao1	558.4c	627b	634.4b	677.2a	29.08	0.01	0.66	0.47
Simpson	0.95	0.94	0.93	0.93	0.01	0.36	0.73	0.83
Shannon	4.22	4.17	4.14	4.26	0.24	0.93	0.73	0.88
PD	40.5c	42.8b	44.7a	44.8a	1.15	0.01	0.35	0.81
**Kingdom: Phylum (%)**								
Bacteria: Firmicutes	73.8a	73.43a	71.27 b	69.37c	0.24	<0.001	0.005	0.007
Bacteria: Bacteroidetes	21.51c	22.09c	23.23b	25.39a	0.24	<0.001	0.005	0.68
Bacteria: Proteobacteria	3.22a	3.06b	2.24c	2.04d	0.025	<0.001	0.36	<0.001
Bacteria: Actinobacteria	0.16	0.16	0.16	0.16	0.005	0.86	0.46	0.62
Bacteria: Spirochaetes	0.14	0.14	0.13	0.15	0.01	0.83	0.38	0.83
Bacteria: Tenericutes	0.06c	0.05c	0.12b	0.17a	0.01	<.001	0.005	0.07
Bacteria: Cyanobacteria	0.15	0.16	0.16	0.15	0.005	0.94	0.39	0.82
Bacteria: Verrucomicrobia	0.09b	0.17b	2.15a	2.14a	0.03	<0.001	0.09	<0.001
Archaea: Euryarchaeota	0.87a	0.74b	0.54c	0.45d	0.013	<0.001	<0.001	<0.001

a, b, c, d: Values in the same row with different letter superscripts mean significant differences (*p* < 0.05). Data are presented as the mean ± SEM (*n* = 6). Abbreviations: L, linear; Q, quadratic; C, cubic.

**Table 9 animals-11-00817-t009:** Effect of dietary tryptophan levels on the microbial composition in the caecum of weaned piglets at the genus level.

Items	Dietary Tryptophan Levels				SEM	*p*-Value		
	0.14%	0.21%	0.28%	0.35%		L	Q	C
**Family: genus (%)**								
V1: unculture	44.20a	34.80b	29.47c	23.38d	0.960	<0.001	0.1	0.28
V1: *Phascolarctobacterium*	1.13b	1.17b	1.33a	1.35a	0.040	<0.001	0.87	0.11
V1: *Succiniclasticum*	0.17b	0.18a	0.18a	0.18a	0.004	0.02	0.08	0.41
V1: *Acidaminococcus*	1.30	1.40	1.42	1.42	0.040	0.05	0.18	0.79
V1: *Mitsuokella*	2.26a	1.74b	1.74b	1.57c	0.031	<0.001	<0.001	<0.001
V1: *Anaerovibrio*	0.94a	0.85a	0.74b	0.64c	0.720	<0.001	<0.001	<0.001
V1: *Dialister*	0.12b	0.13b	0.14b	0.17a	0.008	<0.001	0.39	0.29
V1: *Megasphaera*	3.46a	2.57b	2.71b	2.42b	0.150	<0.001	0.07	0.05
L1: *Blautia*	0.12d	0.32c	0.6b	0.94a	0.090	<0.001	0.46	0.97
L1: *Roseburia*	0.05d	0.12c	0.2b	0.26a	0.010	<0.001	0.59	0.67
L1: unculture	2.21c	3.07b	3.63c	3.93a	0.110	<0.001	0.03	0.90
L1: *Lachnospira*	0.11b	0.17a	0.17a	0.18a	0.070	<0.001	0.02	0.09
L1: *Coprococcus*	0.27d	0.5c	0.61b	0.8a	0.020	<0.001	0.68	0.12
R: *Faecalibacterium*	0.69c	1.07b	1.44a	1.61a	0.060	<0.001	0.11	0.54
R: *Ruminococcus*	0.2c	0.62b	0.92a	0.94a	0.030	<0.001	<0.001	0.30
R: *Oscillospira*	0.18c	0.24b	0.26b	0.31a	0.010	<0.001	0.73	0.38
R: unculture	2.62d	2.97c	4.03b	4.37a	0.050	<0.001	0.89	<0.001
C1: *Clostridium*	2.78	2.56	2.60	2.70	0.120	0.68	0.19	0.68
C2: unculture	3.64	3.49	3.64	3.74	0.140	0.40	0.18	0.56
P1: unculture	0.04	0.03	0.03	0.03	0.006	0.05	0.54	0.90
S1: *Streptococcus*	3.44b	4.82a	5.00a	5.31a	0.190	<0.001	0.01	0.13
L2: *Lactobacillus*	1.77d	7.27c	8.72b	11.74a	0.100	<0.001	<0.001	<0.001
T: *Turicibacter*	1.94a	1.74b	1.54c	1.25d	0.030	<0.001	0.24	0.54
E1: p-75-a5	0.03	0.04	0.03	0.04	0.006	0.33	0.87	0.13
E1: *Allobaculum*	0.03	0.02	0.03	0.03	0.004	0.76	0.81	0.36
B1: *Bacteroides*	0.06	0.06	0.06	0.07	0.009	0.42	0.81	0.92
P2: *Prevotella*	12.05a	11.33b	10.24c	10.38d	0.780	0.01	0.03	0.20
B2: unculture	2.34b	2.44b	2.78a	2.97a	0.110	<0.001	0.68	0.43
M1: unculture	3.78c	4.78b	5.41a	5.56a	0.060	<0.001	<0.001	0.66
RF16: unculture	0.22	0.24	0.22	0.23	0.080	0.48	0.64	0.12
P3: *Parabacteroides*	0.86	0.85	0.87	0.83	0.700	0.67	0.32	0.19
P4: *Paraprevotella*	0.18	0.19	0.19	0.2	0.010	0.21	0.92	0.89
P4: CF231	0.06b	0.07a	0.07a	0.07a	0.003	0.01	0.06	0.37
E2: unculture	0.59a	0.57a	0.39b	0.36b	0.02	<0.001	0.8	2
S2: *Succinivibrio*	1.74a	1.58b	1.38c	1.24d	0.02	<0.001	0.55	0.3
R2: unculture	0.06a	0.06a	0.04b	0.04b	0.04	<0.001	0.98	0.04
A: *Sutterella*	0.36a	0.4a	0.2b	0.15b	0.02	<0.001	0.03	<0.001
D: *Desulfovibrio*	0.45a	0.44a	0.23b	0.24b	0.01	<0.001	0.31	<0.001
B3: *Bifidobacterium*	0.11	0.11	0.11	0.10	0.005	0.32	0.26	0.61
C2: unculture	0.05	0.05	0.05	0.06	0.004	0.32	0.04	1.00
S3: *Treponema*	0.06c	0.05c	0.12b	0.17a	0.01	<0.001	0.005	0.07
RF39: unculture:	0.14	0.14	0.13	0.15	0.01	0.83	0.38	0.83
P5: *Planktothrix*	0.15	0.16	0.16	0.15	0.005	0.94	0.39	0.82
RFP12: unculture	0.02c	0.04c	0.84b	0.77a	0.02	<0.001	0.03	<0.001
V2: *Akkermansia*	0.07b	0.12b	1.31a	1.34a	0.02	<0.001	0.69	<0.001
M2: *Methanobrevibacter*	0.87a	0.74b	0.54c	0.45d	0.013	<0.001	<0.001	<0.001

a, b, c, d: Values in the same row with different letter superscripts mean significant differences (*p* < 0.05). Data are presented as the mean ± SEM (*n* = 6). Abbreviations: A, Alcaligenaceae; B1: Bacteroidaceae; B2: Bacteroidales; B3: Bifidobacteriaceae; C1, Clostridiaceae; C2: Coriobacteriaceae; D: Desulfovibrionaceae; E1: Erysipelotrichaceae; E2: Enterobacteriaceae; L1, Lachnospiraceae; L2: Lactobacillaceae; M1:Muribaculaceae; M2, Methanobacteriaceae; R1: Ruminococcaceae; R2, Rhodocyclaceae; P1, Peptostreptococcaceae; P2: Prevotellaceae; P3: Porphyromonadaceae; P4: Paraprevotellaceae; P5, Phormidiaceae; S1, Streptococcaceae; S2, Succinivibrionaceae; S3: Spirochaetaceae; T: Turicibacteraceae; V1: Veillonellaceae; V2: Verrucomicrobiaceae; L, linear; Q, quadratic; C, cubic.

## Data Availability

The data presented in this study are available on request from the corresponding author. The data are not publicly available due to the regulations of National Natural Science Foundation of China.

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
