# Peer review of "Dietary Tryptophan Levels Impact Growth Performance and Intestinal Microbial Ecology in Weaned Piglets via Tryptophan Metabolites and Intestinal Antimicrobial Peptides"

_animals, 2021, doi:10.3390/ani11030817_

Round 1

Reviewer 1 Report

The topic of the article is supplementation of Tryptophan to weaned piglets and evaluation a lot of different factors including growth performance, intestinal microbial ecology, metabolites and antimicrobial peptides, which is interesting and actual topic. There are described really a lot of different parameters.

I have following questions and recommendations:

Abstract

Do not use the abbreviations (HT, mTOR)

Keywords

The words from the title are repeated (tryptophan, growth performance, intestinal microbial ecology) should be different.

Abbreviations

In the text is lot of different abbreviations. They must be explain when are used for the first time. It is not always done (line 157 – BD-2, mTORC; line 202 – SEM; line 274 – IDO, line 314 ANG-1, FABP-2, p-mTOR, p-AKT….). It would useful to show summary of all abbreviations also separately.

Material and Methods

2.1. Animal use and care – this part is redundant, because animals are described in the next part and first part (Lines 61-64) do not describe information about animal use and care.

Composition of the ration. You write “DE is calculated and others are measured values” Is it really true that you measured the composition of 4 rations and in all rations were precisely the same concentrations of all nutrients? Did you check the content of tryptophan? The values look like calculated and not measured values. Mainly important is if you check the concentration of tryptophan in individual experimental rations.

Methods for measurement of following parameters are not described: GH, cortisol, insulin, melatonin, TPH, IDO, 5-HT, IL-17, IL-22. Please add these methods.

Results

The description of the results is quite long and mostly are only described the results which are in tables including p-values, which is redundant. Please summarize the most important results in the text without showing p-value, everybody could check the tables. The description of the results should be substantially shorten.

Table 3 – There are values of the weight for thyroid, thymus, liver, spleen, pancreas, longissimus dorsi muscle and then there are indexes for the same organs. It is redundant to show both values (weight and index) for all organs. Moreover it is clear that if the body weight of piglets in experimental groups is much higher than in control, that also organs weights will be higher. I recommend to show only indexes.   

Line 238 – abbreviation for daily feed intake is not correct, should ADFI not AFI

Line 335 – the description of the figure is missing

Author Response

dear reviewer

We have revised the manuscrips, please see attachments file. thanks.

Zhiru

Reviewer 2 Report

Diet of piglets prior to weaning should be specified.

Conclusions should include an indication of the value of the results, for example, by suggesting a modified inclusion rate of tryptophan in the diet.

A few spots are confusingly worded.

Author Response

Dear reviewer

We have revised the manuscripts, please see attachents file.

Zhiru

Round 2

Reviewer 1 Report

I have no more sugestion.